# Chronic Kidney Disease—Mineral and Bone Disorder (CKD-MBD), from Bench to Bedside

**Kosaku Nitta** [1,*], **Norio Hanafusa** [2], **Kenichi Akiyama** [1], **Yuki Kawaguchi** [2] **and Ken Tsuchiya** [2]

[1] Department of Nephrology, Tokyo Women's Medical University, Tokyo 162-8666, Japan
[2] Department of Blood Purification, Tokyo Women's Medical University, Tokyo 162-8666, Japan
* Correspondence: knitta@twmu.ac.jp

**Abstract:** Chronic kidney disease—mineral and bone disorder (CKD-MBD) is a systemic disorder that increases the risk of morbidity and mortality in dialysis patients. CKD-MBD is highly prevalent in dialysis patients, and appropriate treatment is important for improving their outcomes. Inorganic phosphate, fibroblast growth factor 23, parathyroid hormone, and calciprotein particles are markers for critical components and effectors of CKD-MBD, and higher circulating levels of these markers are linked to cardiovascular diseases. In this short review, we focus on the pathogenesis and management of CKD-MBD in CKD patients, especially those on dialysis therapy, and discuss the prospects for improving the management in CKD patients, including those on dialysis.

**Keywords:** CKD-MBD; hemodialysis; phosphate; cakciprotein particle; vascular calcification; mortality





## 1. Introduction

Abnormalities of bone and mineral metabolism in patients with chronic kidney disease (CKD) were previously regarded as a disorder of the bones and parathyroid glands [1]. However, these abnormalities are now recognized as a systemic disorder that affects a wide variety of systems, including cardiovascular (CV) organs, and is referred to as CKD—mineral and bone disorder (MBD) [2,3]. Abnormal serum levels of phosphate (P), calcium (Ca), and parathyroid hormone (PTH) are associated with increased risk of morbidity and mortality in hemodialysis (HD) patients [4], and various evidence has confirmed that hyperphosphatemia is closely related to higher risk of CV events and death [5]. A growing body of evidence has recently shown that increased serum fibroblast growth factor (FGF)-23 levels are associated with left ventricular hypertrophy, vascular calcification (VC), infection, anemia, and inflammation in HD patients [6]. Because serum levels of PTH and FGF-23 are increased in response to phosphate loading, lowering serum P and reducing P loading is crucial in the management of CKD-MBD in HD patients [7].

CKD-MBD involves biochemical abnormalities, bone disorders, and VC, all of which can cause CV events, bone fractures, and other serious complications, ultimately leading to death in CKD patients, including those on dialysis [2,3]. Notably, the concept of CKD-MBD continues to develop and the scope of this disorder is expanding to encompass a wide range of diseases, due to advances in our understanding of the pathogenesis of CKD-MBD.

## 2. P Imbalance and CKD-MBD

The control of serum P is recognized to be important in the treatment of CKD-MBD because of both the role of P overload in the development of the disorder and the independent association between hyperphosphatemia and CV disease (CVD) [8]. Good management of P metabolism likely reduces the risk of VC [9], secondary hyperparathyroidism (SHPT) [10], and decreased FGF-23 production [11], thus slowing the progression of CKD-MBD and reducing CV mortality risk. However, an obstacle to effective P control

for the treatment of CKD-MBD is the lack of P management strategies that can be used to maintain P concentrations within the normal range of <5.5 mg/dL, in dialysis patients [3].

Phosphate binders (PBs) have recently been used as a pharmacological treatment for hyperphosphatemia. However, PBs bind only a portion of dietary phosphate and require patients to take many pills with meals [12,13]. In addition, proper adherence to PBs is challenging [14]. The effectiveness of PBs and dietary P restriction are further limited by maladaptive upregulation of P absorption [15,16].

New strategies for P management should take into account the latest understanding of P adsorption, namely, that the paracellular P absorption pathway is the dominant route of intestinal P absorption [17]. A novel P absorption inhibitor, tenapanor, has recently been developed. It directly targets the intestinal sodium/hydrogen exchanger isoform 3 (NHE3), leading to reduced sodium absorption [18]. In clinical trials, tenapanor has also been shown to reduce serum P concentrations and to be generally well tolerated [19,20]. It may offer a novel treatment approach for CKD-MBD.

## 3. CKD-MBD: As a Risk Factor for CV Mortality

CKD-MBD is a common comorbidity and a main cause of CV mortality in patients on dialysis [2]. Along with declines in kidney function, progressive disruption in Ca and P metabolism is associated with abnormalities in circulating hormone concentrations such as PTH and FGF-23 and decreases in calcitriol (Figure 1) [21].

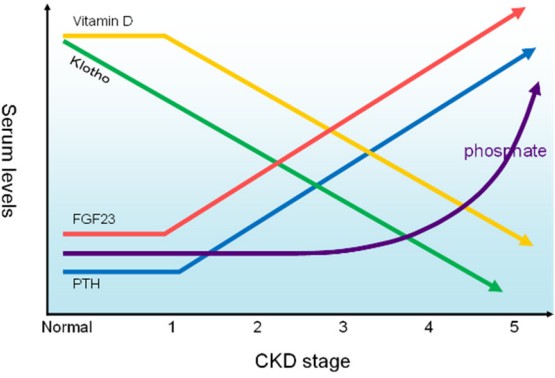

**Figure 1.** Changes in biochemical markers of MBD during the progression of CKD. Reproduced with permission from John, GB et al., Am J Kidney Dis, published by Elsevier, 2011.

Various components of CKD-MBD such as hyperphosphatemia, VC, and elevated FGF-23 concentrations are known to be significantly associated with increased CV morbidity and mortality [3]. CVD accounted for 45% of deaths among Japanese dialysis patients in 2021 [22]. Traditional risk factors for CV mortality, such as hypertension and diabetes, do not explain the higher CV morbidity and mortality in dialysis patients [23], and established treatment strategies for these risk factors have not seen significant advancements. Better control of P may be necessary to improve clinical outcomes and quality of life in dialysis patients and reduce mortality risk [24].

Therapeutic approaches for CKD-MBD that may decrease CV mortality include improving dialysis modalities [25], decreasing inflammation [26], and better managing serum P concentrations [27]. In particular, hyperphosphatemia is a major modifiable target [28]. An abnormal P concentration has been identified as an independent risk factor for CV morbidity and mortality in dialysis patients [29]. A linear correlation has been found between progression of coronary calcification and increasing serum P concentrations [30].

## 4. The Pathogenesis and Associations of VC with CKD-MBD

Among the three components of CKD-MBD, VC has recently been in the spotlight, with various research seeking to clarify its pathogenesis in CKD patients [31]. VC is a common complication in CKD patients and is associated with increased CV morbidity

and mortality [32,33]. VC was previously regarded as a passive and degenerative process of Ca deposition in the vessel wall, but it is now recognized as an actively regulated cellular process [34]. Table 1 shows the risk factors for VC in dialysis patients. Over the past decade, basic research has revealed that VC is mediated by complex cellular mechanisms including transdifferentiation of vascular smooth muscle cells (VSMCs) into osteoblast-like cells, apoptosis of VSMCs, degradation of the extracellular matrix, formation and release of calcifying matrix vesicles, and formation and maturation of calciprotein particles (CPPs) [35]. Among these factors, CPPs have attracted research interest in the fields of nephrology and CKD-MBD and are now suggested to be a critical mediator of VC [36,37]. CPPs contain Ca, inorganic P, fetuin-A, and other proteins and increase in response to an increasing P and Ca burden, inducing inflammatory responses in leukocytes, monocytes, renal tubular cells, and VSMCs (Figure 2) [37,38]. When VSMCs are exposed to high CPP conditions, CPPs enter the intracellular space through scavenger receptor A or act on the cells through certain Toll-like receptors and induce intracellular Ca overload, resulting in apoptosis, altered autophagy, and calcification of the extracellular matrix [39,40]. CPPs are now considered one of the strong drivers for uremic VC.

**Table 1.** Risk factors for vascular calcification.

| Clinical | <ul><li>Age</li><li>Duration of Dialysis</li><li>Kidney Function/Uremia</li><li>Diabetes</li><li>Known Coronary Artery Disease</li><li>Abnormal Bone</li></ul> |
|---|---|
| Biochemical | <ul><li>Hyperphosphatemia</li><li>Hypercalcemia</li><li>High Parathyroid Hormone</li><li>Low Fetuin-A</li><li>Increased Aldosterone</li><li>Oxidative Stress</li><li>Low Pyrophosphate</li><li>Decreased MGP</li><li>Decreased BMP-7</li></ul> |
| Medications | <ul><li>Calcium-Containing Phosphate Binders</li><li>High-Dose Vitamin D</li><li>Coumadin (Decreases Active MGP)</li></ul> |

MGP, matrix GIa protein; BMP, bone morphogenic protein.

For VC, another important consideration is imbalance between inducers and inhibitors of calcification. In CKD, inducers of calcification such as P and Ca loading are accumulated, while calcification inhibitors such as fetuin-A, pyrophosphate, and magnesium (Mg) are decreased in the circulation, thereby accelerating the VC process [41].

Unfortunately, treatment of VC remains challenging in the clinical settings. However, basic studies have provided some ideas for preventing progression of VC in CKD patients. In an experimental study, dietary P restriction or treatment with PBs prevented or halted the progression of uremic VC [42]. Oxidative stress, which is high in CKD patients, including the dialysis population, plays an important role in the pathogenesis of VC, and treatment with antioxidants slows the progression of VC in uremic rodents [43]. Mg ions, which were recently reassessed as a potentially attractive therapeutic option in CKD, inhibited the formation and maturation of CPPs and prevented inflammation and VC in CKD [44]. VC is considered a largely irreversible biochemical phenomenon, so it is very difficult to achieve regression of VC once it has formed in the blood vessel wall, although some therapeutic interventions have been reported to reverse VC [45]. Therefore, at present, prevention of VC is a reasonable and feasible strategy in CKD patients, including dialysis population.

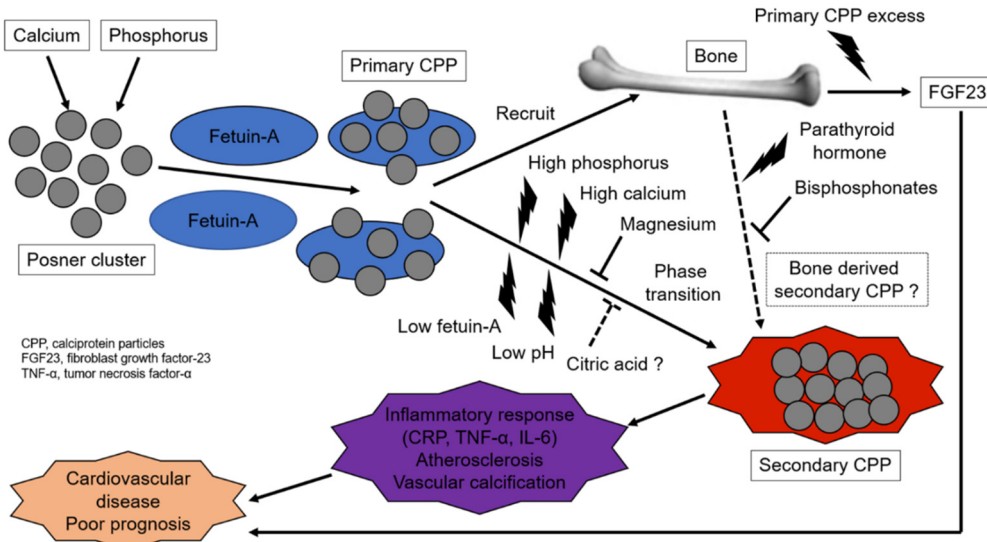

**Figure 2.** Role of CPP in CKD-MBD.

## 5. Biomarkers of CKD-MBD Assessment

With progression of CKD, the glomerular filtration rate is reduced, leading to P retention [46]. In earlier CKD stages, PTH and FGF-23 concentrations steadily increase, while P concentrations are stable [22]. Abnormalities in mineral metabolism (e.g., SHPT, decreased vitamin D) are seen in CKD patients before hyperphosphatemia develops [47]. Increased PTH [48,49], increased FGF-23 [50], hypocalcemia [51], and low vitamin D concentrations [52] have all been found to be associated with increased risk of CVD. Inhibition of calcitriol production due to increased FGF-23 [53] has been associated with coronary artery calcification [54] and myocardial fibrosis [55].

In CKD stages 4 and 5, P concentrations begin to increase despite the elevations in FGF-23 and PTH levels, indicating that compensatory mechanisms are no longer sufficient to maintain P balance and prevent hyperphosphatemia [22]. CKD patients with poorly managed P metabolism have almost 30% greater mortality risk than those who achieve and continuously maintain normal P concentration [5]. In addition, hyperphosphatemia is directly linked to hypertension, a major CVD risk factor present in up to 90% of CKD patients [56]. Therefore, P control has the potential to be a major therapeutic target for reducing mortality in CKD patients.

FGF-23 and PTH concentrations are key regulators of P metabolism and their concentrations increase in response to P retention [57,58]. Elevated FGF-23 concentrations are the earliest mineral metabolism abnormality observed in CKD patients [22]. P retention stimulates progressive increases in FGF-23 concentrations [53]., which directly target the heart to promote left ventricular hypertrophy and congestive heart failure [59]. Higher FGF-23 concentrations are independently associated with a greater risk of CV events, particularly congestive heart failure [50]. Elevated PTH concentrations are associated with various negative effects including increased bone resorption [60], decreased cardiac index and mean arterial pressure via impaired myocardial energy production [61], and genesis of cardiac fibrosis [62].

## 6. Management of CKD-MBD

CKD-MBD is a multifactorial disease. Despite of the great efforts to improve the management of patients with CKD-MBD, several unmet needs still remain. The management of CKD-MBD is based on several strategies to prevent the adverse complications associated with SHPT. The current KDIGO guideline recommends treatment of SHPT based on repeated measures of the biochemical markers including serum P, Ca and PTH [63]. Abnormal P/Ca metabolism is a major characteristic of CKD-MBD, which is associated

with clinically significant VC and increased mortality risk [3]. In Japan, the target range of serum Ca concentrations was decided according to the results of an analysis of data from the Japanese Society for Dialysis Therapy (JSDT) patient registry, which collected data on patients treated in accordance with the 2008 JSDT guideline [64].

After the 2008 JSDT guideline was issued [64], awareness of CKD-MBD increased in Japan, and new drugs have been considered for the treatment of CKD-MBD, such as cinacalcet hydrochloride, lanthanum carbonate and iron-containing PBs.

Based on the analysis of data from 128,125 dialysis patients who could be monitored from the end of 2006 to the end of 2009 [4], the target range for serum P range was set at 3.5–6.0 mg/dL. The dialysis serum P range was set at 3.5–5.5 mg/dL in the KDOQI clinical practice guideline, while the 2009 CKD-MBD KDIGO guidelines [3] recommended that the serum P concentration be lowered if higher than the reference level.

The target range for serum Ca concentration is 8.4–10.0 mg/dL in the JSDT guideline [64]. The KDOQI guideline states that serum Ca levels should be 8.4–9.5 mg/dL, while KDIGO guidelines states that the target should be within the normal range [3].

The JSDT guideline clearly recommends that control of serum P should be given the highest priority, followed by control of serum Ca, and then PTH. Previous studies have shown that prognosis is improved more by appropriate control of serum P and Ca levels compared with control of PTH alone [65–67]. In Japan, the serum levels of P and corrected Ca should be controlled first, and then the dose of vitamin D receptor activators (VDRAs) or cinacalcet hydrochloride should be adjusted to maintain serum PTH levels within the target range (60–240 pm/mL).

Various PBs are available and can be used when tailored to individual patients (Table 2). When PBs are prescribed, patient adherence must be confirmed. Moreover, it is important to bear in mind that certain drugs are more effective when taken at specific times. For instance, sevelamer hydrochloride, bixalomer and sucroferric oxyhydrate (SFOH) should be taken before a meal, and $CaCO_3$, lanthanum carbonate and ferric citrate hydrate (FCH) should be taken immediately after a meal.

Drugs for CKD-MBD-related drugs should be considered from perspective of not only serum P/Ca control but also prognosis. A number of observational cohort studies of dialysis patients indicate that administration of VDRA is associated with lower all-cause and CV mortality risk, independent of control of serum P, Ca, or PTH levels [68–72]. One the options for SHPT treatment is to combine low-dose VDRAs with calcimimetics [73].

Cinacalcet hydrochloride is expected to prevent progression of VC and improve prognosis because it can simultaneously lower serum P/Ca/PTH levels. Analysis of the combined results of four safety survey studies showed that the risk for vascular disease-related hospitalization was reduced in patients taking cinacalcet hydrochloride [74]. Moreover, a large-scale observational study revealed that cinacalcet hydrochloride was associated with reduced risk all-cause and cardiovascular mortality [75]. Recently, the Evaluation of Cinacalcet HCl Therapy to Lower cardioVascular Events (EVOLVE) study, a double-blind randomized controlled study, showed that cinacalcet hydrochloride did not significantly reduce the risk of death or major CV events in dialysis patients with moderate-to-severe SHPT [76].

Isakova et al. reported that the risk of mortality was lower in patients treated with PBs than in those not treated with PBs [77]. An appropriate upper limit for $CaCO_3$ administration is around 3 g/day, considering the importance of avoiding excessive Ca load [78]. Reduction of $CaCO_3$ should be considered when hypercalcemia is likely to occur, there is marked VC, adynamic bone disease is suspected, or the blood PTH level is constantly low [78]. In addition, switching to Ca-free PBs is reasonable.

**Table 2.** Comparison of phosphate binders in chronic kidney disease.

| Phosphate Binder | Pros | Cons |
|---|---|---|
| Calcium-based: calcium acetate calcium carbonate calcium citrate | • Increases calcium and can correct hypocalcemia<br>• Low cost<br>• Moderate pill burden | • Hypercalcemia and/or positive calcium balance<br>• Cardiovascular calcification |
| Sevelamer-based: sevelamer carbonate sevelamer hydrochloride | • No systemic absorption<br>• Potentially less vascular calcification (calcium-free)<br>• Lowers LDL cholesterol<br>• Improvement in metabolic acidosis with carbonate variant | • Adverse GI effects<br>• High pill burden<br>• High cost<br>• Binds fat-soluble vitamins<br>• Metabolic acidosis with the hydrochloride variant |
| Iron-based: sucroferric oxyhydroxide | • Lower pill burden<br>• Minimal systemic absorption, no iron overload<br>• Greater efficacy<br>• Increased GI motility which might be beneficial in constipated and PD patients | • High cost |
| Iron-based: ferric citrate | • Noninferior to sevelamer, well tolerated, beneficial effect on renal anemia | • Systemic absorption with potential for iron overload |
| Lanthanum carbonate | • Twice as potent as calcium and sevelamer | • High cost<br>• Systemic absorption and potential tissue deposition/toxicity<br>• GI intolerance, nausea<br>• Difficult to chew |

GI, gastrointestinal; LDL, low-density lipoprotein; PD, peritoneal dialysis. Reproduced with permission from Rastogi, A. et al., J Ren Nutr, published by Elsevier, 2021.

The tight links among iron deficiency, renal anemia, and CKD-MBD has opened the door to the use of iron-containing PBs with the aim of iron supplementation and subsequent improvement in blood hemoglobin levels, CVD, and survival [79]. Although the efficacy and safety of ferric citrate hydrate (FCH) and sucroferric oxyhydroxide (SFOH) were found to be similar to those of sevelamer, there have been no head-to-head studies with lanthanum carbonate [80,81]. Clinical data for 1 year in a small patient cohort suggested improved adherence with SFOH, and a large randomized controlled trial is now needed to confirm these possible advantages. Cost-effectiveness in comparison with other PBs and the safety of long-term treatment will determine the future use of both FCH and SFOH.

### 7. Conclusions

CKD-MBD is highly prevalent complication in CKD patients, including those on dialysis, and contribute to increased risk of CV mortality. It is important to understand that P retention triggers increases in CPPs and serum FGF-23 and PTH, but these increases may be associated with CVD. Dietary restriction of P is complicated by inorganic P in food additives, and the use of many medications makes control of P difficult for dialysis patients with CKD-MBD. Dialysis can only remove ~900 to 1200 mg of P per day. PBs, including iron-containing ones, constitute the major pharmacological treatment for hyperphosphatemia and VC. When PBs are insufficient to maintain a normal P concentration, combination of low-dose VDRAs with cinacalcet hydrochloride is useful for treating CKD-MBD in dialysis patients with SHPT. Tenapanor is a novel P absorption inhibitor that acts on the primary absorption pathway, providing a new approach to treating hyperphosphatemia. Further research is necessary to determine whether tenapanor is effective in reducing the CVD risk in dialysis patients.

**Author Contributions:** K.N. wrote the draft of the manuscript and N.H., Y.K., K.A. and K.T. revised the manuscript. All authors have read and agreed to the published version of the manuscript.

**Funding:** This research received no external funding.

**Institutional Review Board Statement:** Not applicable.

**Informed Consent Statement:** Not applicable.

**Data Availability Statement:** Not applicable.

**Acknowledgments:** We would like to express our sincere thanks to all the doctors and medical staff to collect data in Departments of Nephrology and Blood Purification.

**Conflicts of Interest:** The authors have no conflict of interest to declare.

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
