# Peer review of "Chronic Kidney Disease—Mineral and Bone Disorder (CKD-MBD), from Bench to Bedside"

_kidneydial, doi:10.3390/kidneydial3010004_

Round 1
Reviewer 1 Report
This is an interesting review on a very important topic in nephrology.
Overall the review is well organized and presented, but I have several concerns that need to be addressed.
Concerns:
1) English grammar is poor and sometimes prevents clear understanding of the presentation. This needs to be corrected.
2) In line 163, ion-containing PBs should be replace by iron-containing PBs
3) In line 178, the authors are including cinacalcet as a vitamin D receptor activator? Cinacalcet is a calcium receptor agonist (calcimimetic).
4) In line 191 and 192, the authors make the statement that: “The worldwide trend for SHPT treatment is directed toward combination of low-dose VDRAs with calcimimetics” What are the objective data to support such statement?
5) In line 227, please change inorganize for inorganic
6) In line 229, the range of phosphorus removal during dialysis seems too wide: 300 to 1200 mg per day?
7) In line 231, 232, I imagine there is typo for low-dose VDRAs
Author Response
Response to reviewer’s comments
Kidneydial-1927-147
Review Article
Nitta et al. Chronic Kidney Disease – Mineral Bone Disorder (CKD-MBD), from Bench to Bedside
Reviewer: This is an interesting review on a very important topic in nephrology. Overall the review is well organized and presented, but I have several concerns that need to be addressed.
<Response>
Thank you so much for your encouragement. We revised our manuscript according to the comments.
Concerns:
- English grammar is poor and sometimes prevents clear understanding of the presentation.
<Response>
We changed the part of manuscript according to the suggestion of a native English speaker as underlined.
- In line 163, ion-containing PBs should be replace by iron-containing PBs.
<Response>
We changed the word as mentioned.
- In line 178, the authors are including cinacalcet as a vitamin D receptor activator? Cinacalcet is a calcium receptor agonist (calcimimetics)
<Response>
We mentioned “vitamin D receptor activator or cinacalcet hydrochloride” in the sentence.
- In line 191 and 192, the authors make the statement that: “The worldwide trend for SHPT treatment is directed toward combination of low-dose VDRAs with calcimimetics”. What are the objective data to support such statement?
<Response>
We are sorry to mention an inadequate statement. We revised the part as “An option for SHPT treatment is combination of low-dose VDRAs with calcimimetics [72].”
- In line 227, inorganize for inorganic
<Response>
We changed inorganize to inorganic.
- In line 229, the range of phosphorus removal during dialysis seems too wide: 300 to 1200 mg per day?
<Response>
We changed to “900 to 1200 mg per day.
- In line 231, 232, I imagine there is typo for low-dose VDRAs.
<Response>
We changed “ow” to “low”.
Reviewer 2 Report
The article is sometimes repetitive and it lacks pathophysiology explanations especially in the later parts of the article.
Maybe it would be good to rephrase this part of the sentence. "VC is very popular in the CKD patients..."
It is unnecessary to emphasize the obvious: " In animal studies, P loading dose-dependently induced VC and hyperphosphatemia in 115 rodents with CKD ..."
Crossed out word "Meanwhile, Mg ..."
Could be one sentence: "However, increases in PTH and FGF-23 have both been associated with increased risk of CVD [48-50]. Hypocalcemia and low vitamin D concentrations have been associated with increased risk of CVD [51,52]. "
CKD-MKD is a global problem and as such is regulated by each country with their own guidelines. There is no need in mentioning regional recommendations because the above is implied. "In Japan, the target range of serum Ca concentrations were decided according to-the results of an analysis of data from the Japanese Society for Dialysis Therapy (JSDT) patient registry, where patients were treated in accordance with the 2008 JSDT guideline [63]."
Needs revision: "...serum P level should be lowered if high higher than the reference level."
Is it one sentence? "CKD–MBD is highly prevalent complications in the CKD patients including dialysis population, and synergistically contributes to heightened risk of CV morbidity and mortality. Given an understanding that P retention triggers and derives the increase in CPPs and serum FGF-23 and PTH."
Part 6. and Conclusion need thorough revision.
References should be updated, most of them are from 2000-2010.
Author Response
Response to reviewer’s comments
Kidneydial-1927-147
Review Article
Nitta et al. Chronic Kidney Disease – Mineral Bone Disorder (CKD-MBD), from Bench to Bedside
Reviewer: The article is sometimes repetitive and it lacks pathophysiology explanations especially in the latter part of the article.
<Response>
Thank you so much for valuable suggestions. We revised our manuscript according to the comments.
- Maybe it would be good to rephrase this part of the sentence. “VC is very popular in the CKD patients.
<Response>
We rephrased our manuscript as underlined. asked a native English speaker to review our manuscript and blushed up the manuscript as underlined.
- It is unnecessary to emphasize the obvious. “In animal studies, P loading dose-dependently induced VC and hyperphosphatemia in 115 rodents with CKD…”
<Response>
We deleted this sentence.
- Crossed out word “Meanwhile, Mg…”
<Response>
We deleted “Meanwhile”.
- Could be one sentence: “However, increases in PTH and FGF-23 have both been associated with increased risk of CVD [48-50]. Hypocalcemia and low vitamin D concentrations have been associated with increased risk of CVD [51, 52].
<Response>
We separated two words as referred.
- CKD-MBD is a global problem and as such is regulated by each country with their own guidelines. There is no need in mentioning regional recommendations because the above is implled. “In Japan, the target range of serum Ca concentrations were decided according to the results of an analysis of data from the Japanese Society for Dialysis Therapy (JSDT) patient registry, where patients were treated in accordance with the 2008 JSDT guideline [63].
<Response>
We would like to describe Japanese guideline for CKD-MBD and compare it with guidelines of KDOQI and KDIGO.
- Need revision. “..serum P level should be lowered if high higher than the referred level.”
<Response>
We deleted “high”.
- Is it one sentence? “CKD-MBD is highly prevalent complications in the CKD patients including dialysis population, and synergistically contributes to heightened risk of CV morbidity and mortality. Given an understanding that P retention triggers and derives the increase in CPPs and serum FGF-23 and PTH.”
<Response>
We revised the sentence as follows:
“MBD is one of the common complications in the CKD patients including dialysis population, and contributes to heightened risk of CV morbidity and mortality. It is important to understand that P retention triggers and derives the increase in CPPs and serum FGF-23 and PTH.”
- Part 6, and Conclusion need through revision.
<Response>
We changed the part of manuscript according to the suggestion of a native English speaker as underlined.
- References should be updated, most of them are from 2000-2010.
<Response>
We are sorry to inform you that it is hard to change the references because the context will be different. There are still some recent paper from 2017-2020 as references.
Round 2
Reviewer 1 Report
The updated manuscript is markedly improved.
Two minor points still remain:
1) In line 163, ion-containing PBs should be replace by iron-containing PBs
1) In line 226, typo for low-dose VDRAs
Author Response
We have revised our manuscript based on reviewer's comments.
1) line 163; ion-containing PBs have changed to iron-containing PBs.
2) line 226; low-dose
Reviewer 2 Report
Review for international journal should consider multinational view. This review seems to be country oriented (especially in the first part of Management of CKD-MBD section).
120/121 Therefore, at present...
226 low-dose
Author Response
We have revised our manuscript based on reviewer's comments.
1) Management of CKD-MBD; Country oriented description→we added comments from KDIGO conference
2) line 120-121; At present, therefore→Therefore, at present
3) line 226; low-dose